# Evaluating the Contribution of Soybean Rust-Resistant Cultivars to Soybean Production and the Soybean Market in Brazil: A Supply and Demand Model Analysis

**Yuki Ishikawa Ishiwata [1],\* and Jun Furuya [2]**

[1]   Faculty of Life and Environmental Sciences, University of Tsukuba, 1-1-1 Tennodai, Tsukuba, Ibaraki 305-8572, Japan

[2]   Social Sciences Division, Japan International Research Center for Agricultural Science, 1-1 Owashi, Tsukuba, Ibaraki 305-8686, Japan; furuya@affrc.go.jp

\*   Correspondence: ishikawa.yuki.ga@u.tsukuba.ac.jp; Tel.: +81-29-853-4611

**Abstract:** Soybean rust (SBR), caused by *Phakopsora pachyrhizi* (Sydow & Sydow), has become a serious issue in Brazil. As Brazil is one of the largest soybean-producing and exporting countries in the world, a considerable decrease in soybean production due to SBR would have a significant impact on the global soybean market. SBR-resistant cultivars have been developed to prevent a decrease in soybean production. This study was conducted to evaluate the effect of SBR-resistant cultivars on soybean production and the soybean market in Brazil using a supply and demand model. This model consists of functions of yield, cultivated area, exports, and stock changes of soybean and soybean products, demand for soybean products, and price linkages. Five scenarios were simulated to evaluate the economic impact of adopting SBR-resistant cultivars as follows: One without SBR infection, two with serious production losses due to SBR in the south and southeast regions and all the states of Brazil, and two with the adoption of SBR-resistant cultivars in the south and south-east regions and all the states of Brazil. Our simulations suggest that adopting SBR-resistant cultivars reduces the cost of controlling SBR by approximately half and is essential for sustainable soybean production and a stable global soybean market.

**Keywords:** adoption of resistant cultivars; cost saving; fungicide application; production loss; SBR-resistant cultivars

---

## 1. Introduction

Soybean rust (SBR) caused by the harmful fungus, *Phakopsora pachyrhizi* (Sydow & Sydow), is one of the most serious threats to stable soybean production in Brazil [1,2]. Yield losses of over 80% have been reported during times when environmental conditions are optimal for the growth of this pathogen [3,4]. Infected leaves undergo chlorosis, defoliation, and early maturation of the pod, which leads to decrease in yield [2,5]. In Brazil, SBR first occurred in the 2001/2002 crop year, spreading to almost all the soybean-producing states by the 2003/2004 crop year, following which fungicide was applied. However, SBR has proven to be difficult to control and the economic loss in the 2003/2004 crop year was 1.22 billion USD for grain loss and 2.08 billion USD for the cost of applying fungicide [5]. One common strategy of mitigating SBR is by using fungicide, but it increases production costs for farmers [4]. The cost of fungicide accounted for 7.09% of total production cost of soybeans in the 2016/2017 crop year [6], while the average production cost of Brazil in 2013/2014 was approximately twice as high as that in the USA and Argentina [7]. Some studies have suggested that adopting

SBR-resistant cultivars could reduce the cost of fungicide use [4,8–10]. SBR-resistant cultivars have been introduced in Brazil [2], but the highly virulent rust populations have made it difficult to cope with SBR by only one genotype of cultivar [2]. Therefore, SBR-resistant cultivars corresponding to several types of rust population have been developed [2,9,10]. By adopting these cultivars, the cost of applying fungicide can be reduced [2,5].

In 2013, 82 million metric tons of soybeans were produced, making Brazil the second largest soybean-producing country [11], second only to the USA in terms of exports [11]. Soybeans are rich in oil and protein and they are crushed and divided to produce soybean oil and soybean cakes. Soybean oil is not only edible, but it also has industrial uses, such as in biodiesel. With the increasing demand for meat [12], soybean cakes have become a significant source of animal feed. In 2013, Brazil's share in global exports of soybean oil and cakes stood at 12% and 21%, respectively [11]. Hence, the production loss of soybeans caused by SBR can have a huge impact on the global soybean market.

As SBR is affected by environmental conditions such as temperature and precipitation, an epidemic can be influenced by climate change [13]. At the same time, climate change can also affect the yield of soybean. However, different models predicting the effect of climate change on soybean yield have contradicting results [14], with some predicting an increase [15], while others a decrease in yield [16–19]. Thus, the effect of climate change on soybean yield is still ambiguous and there are limited studies on this topic. Alves et al. [13] predicted the future severity of SBR in Brazil with temperature and precipitation as explanatory variables. The study predicted the possibility of a severe SBR outbreak across Brazil given the current situation and emphasized that it would spread to the south (Paraná, Rio Grande do Sul, and Santa Catarina) and southeast (Minas Gerais and São Paulo) of Brazil, which are the main soybean producing areas, with 39.9% production in the 2017/2018 crop year [20]. Therefore, if there is an SBR epidemic in the south and southeast Brazil, soybean production throughout Brazil could fall significantly. The impact of SBR on soybean yield has been reported in the Paraná state, which has one of the highest yields in south Brazil [21]. Rodrigues et al. [21] indicated that soybean yield can be adequately maintained with SBR-resistant cultivars and appropriate use of fungicide. However, fungicide becomes ineffective over time as fungicide-resistant races of SBR appear [5]. With fungicide becoming ineffective against SBR, leading to an increase in the loss of soybean production, what will be the extent of this loss in the south and southeast regions and all the states of Brazil? Furthermore, to what extent will the soybean market be affected? If these cultivars spread throughout Brazil when the effect on soybean production due to SBR is the worst, to what extent will soybean production and the market recover?

Several studies have been conducted to derive economic predictions of soybean production in Brazil [22–27], which has been projected to remain one of the main soybean-producing and exporting countries in the world [22–25,27]. To investigate the economic impact of SBR-resistant cultivars on soybean production, Brazil should be treated separately, because the development of SBR depends on environmental conditions, making the severity of its impact on soybean production vary among countries [2,4]. The analysis for Brazil to determine the soybean harvested areas is based on the model by Moraes [26]. However, most models of soybean production across the world consist not only of area, but also yield, exports, and other components [22–24] which have not been dealt with by the model of Moraes [26]. As Brazil is a main soybean producing and exporting country [11], the impact of soybean production on the soybean market can be taken into account by considering these variables in the model. In this study, a supply and demand model was applied in each state or region of Brazil to evaluate the economic impact of SBR-resistant cultivars on soybean production. SBR is affected by climate conditions that vary between states. The analysis becomes more realistic upon examining these effects with state-level data.

The purpose of this study is to investigate the effects of SBR and the adoption of SBR-resistant cultivars on soybean production and the soybean market in the south and southeast regions and all the states of Brazil. We used the following five scenarios: the base scenario is the current situation, Scenario 1, which assumes that soybean production is not infected by SBR; Scenario 2, which assumes that

fungicide becomes ineffective and soybean production gradually decreases due to SBR in south and southeast Brazil; Scenario 3, which assumes the same case as in Scenario 2, but for entire Brazil; Scenario 4, which assumes the adoption of SBR-resistant cultivars in south and southeast Brazil; and Scenario 5, which assumes the same case as in Scenario 4, but for entire Brazil. We evaluated the economic impacts of SBR on soybean production and the adoption of SBR-resistant cultivars by comparing Scenarios 3 and 5.

## 2. Materials and Methods

### 2.1. Supply and Demand Model of Soybeans and Soybean Products

To investigate the effects of damage due to SBR on the supply and demand of soybeans and soybean products, we used a supply and demand model based on the models of Koizumi and Ohga [24] and Hung et al. [28]. We modified the original model and used state-level data. Because the data, in some states in the north and northeast region, were not available, we summed the data of the north (Rondônia, Roraima, and Tocantins) and northeast (Bahia, Maranhão, and Piauí) regions as north and northeast.

The supply and demand model for soybeans in Brazil consists of 29 structural equations and five identities, specified as follows:

Yield ($Y$) function of soybeans in each state or region (10 functions):

$$Y_{i,t} = \alpha_{yi} + \beta_{yi} T + \varepsilon_{yi,t}, \tag{1}$$

where $\alpha$ and $\beta$ are the parameters and the subscripts $i$ and $t$ are state and year, respectively, $T$ is the time trend, and $\varepsilon$ is an error term.

Planted area of soybeans in each state or region (10 functions):

$$A_{i,t} = \alpha_{ai} + \beta_{a1i}A_{i,t-1} + \beta_{a2i}sbRFP_{i,t-1} + \beta_{a3i}maRFP_{i,t-1} + \varepsilon_{ai,t}, \tag{2}$$

where *sbRFP* and *maRFP* are the real farm prices of soybean and maize, defined as *sbFP*/(*CPI*/100) and *maFP*/(*CPI*/100). *sbFP* and *maFP* are the farm prices of soybean and maize. *CPI* is the consumer price index (2010 = 100).

Country level production (*sbQ*):

$$sbQ_t = \sum_i Y_{i,t}A_{i,t} - bal, \tag{3}$$

where *bal* is the adjustment factor.

The export (*sbEX*) function:

$$sbEX_t = \alpha_e + \beta_{e1} sbQ_t + \beta_{e2}sbRWP_t + \beta_{e3}sbRFP_t + \varepsilon_{te}. \tag{4}$$

where *sbRWP* is the real-world price of soybeans (USD/metric ton) defined as *sbWP*/(*CPI*/100). Here, *sbWP* is the world price of soybeans in US dollars, represented by US No. 2, yellow meal, CIF Rotterdam.

The stock change (*sbSTC*) function:

$$sbSTC_t = \alpha_s + \beta_{s1} (sbQ_t - sbQ_{t-1}) + \beta_{s2}(sbRFP_t - sbRFP_{t-1}) + \varepsilon_{st}, \tag{5}$$

where *sbSTC* is the annual change of stock, i.e., ending stock minus beginning stock.

Soybean supply identity:

$$sbPR_{t=} = sbQ_t + sbIM_t - sbEX_t - sbSTC_t - (sbFO_t + sbSE_t + sbFE_t), \tag{6}$$



where *sbPR*, *sbIM*, *sbFO*, *sbSE*, and *sbFE* are the quantities of soybean processed, imported, used as food, seed, and feed for livestock.

Soybean processing identity:

$$sbPR_t = oilQ_t + cakQ_t + LO_t, \tag{7}$$

where *oilQ* is the quantity of soybean oil production, *cakQ* is the quantity of soybean cake production, and *LO* is the adjustment factor.

Export function of soybean oil (*oilEX*):

$$oilEX_t = \alpha_{oe} + \beta_{oe1}oilQ_t + \beta_{oe2}\frac{oilWP_t}{sbFP_t} + \varepsilon_{oet}, \tag{8}$$

where *oilEX* is the quantity of soybean oil exports and *oilWP* is the world price of soybean oil in US dollars, represented by the Dutch crude degummed, FOB NW Europe.

Export function of soybean cake (*cakEX*):

$$cakEX_t = \alpha_{ce} + \beta_{ce1}cakQ_t + \beta_{ce2}cakRWP_t + \beta_{ce3}sbRFP_t + \varepsilon_{cet}, \tag{9}$$

where *cakEX* is the quantity of soybean cake exports, *cakRWP* is the real-world price of soybean cake (USD/metric ton) defined as *cakWP*/(CPI/100). Here, *cakWP* is the world price of soybean cake in US dollars, represented by the Brazilian pellets 48% protein, CIF Rotterdam.

Demand function of soybean edible oil:

$$oilED_t = \alpha_{of} + \beta_{of1}oilRWP_t + \beta_{of2}cotRFP_t + \beta_{of3}GP_t + \varepsilon_{oft}, \tag{10}$$

where *oilED* is the per-capita consumption of soybean oil, *cotRFP* is the real farm price of cottonseed oil defined as *cotFP*/(CPI/100), and *GP* is defined as per capita real gross domestic product (*GDP*) divided by *POP*. Here, the real *GDP* is *GDP* converted to 2010 constant international dollars using purchasing power parity rates.

Demand function of soybean oil for biodiesel:

$$oilBD_t = \alpha_{bd} + \beta_{bd1}sbRFP + \beta_{bd2}croRWP_t + \varepsilon_{bdt}, \tag{11}$$

where *oilBD* is the quantity of soybean biodiesel oil and *croRWP* is the real-world price of crude oil defined as *croWP*/(CPI/100). Here, *croWP* is the world price of crude oil, represented by the average spot price of Brent, Dubai, and West Texas Intermediate, equally weighed.

Demand function of soybean cake feed:

$$cakFE_t = \alpha_{cd} + \beta_{cd1}sbRFP_t + \beta_{cd2}maRFP_t + \beta_{cd3}chQ_t + \varepsilon_{cdt}, \tag{12}$$

where *cakFE* is the quantity of soybean cake for feed and *chQ* is chicken production (metric ton). Chickens, swine, and cattle have been reared mainly in Brazil, and their productions are strongly correlated ($p < 0.001$). When multiple regression is applied to their production, it will be affected by multicollinearity. To avoid the effect of this problem, we used chicken production as a representative of livestock in Brazil.

Soybean oil identity:

$$oilQ_t = oilED_t + oilBD_t + oilEX_t + oilSTC_t - oilIM_t, \tag{13}$$

where *oilIM* is the imports of soybean oil and *oilSTC* is the stock change in soybean oil.



Soybean cake identity:

$$cakQ_t = cakFE_t + cakEX_t + cakSTC_t - cakIM_t, \tag{14}$$

where *cakSTC* is the stock change in soybean oil and *cakIM* is the imports of soybean oil.

The farm price is linked to the world prices of soybeans and soybean oil and cake. Price linkage functions are as follows:

$$sbWP_t = \alpha_{sb} + \beta_{sb} FP_t + \varepsilon_{sbt} \tag{15}$$

$$oilWP_t = \alpha_o + \beta_o FP_t + \varepsilon_{ot}, \tag{16}$$

$$cakWP_t = \alpha_c + \beta_c FP_t + \varepsilon_{ct}, \tag{17}$$

Farm price is the equilibrium price when the quantity demanded is equal to the quantity supplied.

Figure 1 shows the flowchart of supply and demand for soybean in Brazil in the econometric model. Total production is the sum of production in all the states. Domestic utilization of soybeans is defined as the sum of feed, seed, food, and other forms of utilization. Supply for soybeans is influenced by total production, stock changes, domestic utilization, and the volumes of exports and imports of soybeans. Processing of soybeans is divided into soybean oil and cake production. Supply for soybean oil is influenced by the volumes of exports, imports, and stock changes of soybean oil. Demand for soybean oil is classified into edible soybean oil and biodiesel. The demand for edible soybean oil is affected by population, *GDP*, and domestic farm price of soybeans and cotton seed. The demand for soybean oil for biodiesel is affected by domestic farm price of soybeans and domestic farm price and world price of crude oil. The production of soybean cakes is influenced by the volume of exports and imports and stock changes of soybean cake. The demand for soybean cakes is affected by domestic farm price of soybeans and maize and chicken production. Farm prices are determined when supply and demand are in equilibrium. The movements of soybean oil and cake prices are reflected by farm price through the price linkage functions. The farm price affects the next year's supply. In this sector, import of soybeans and soybean oil and cakes, stock changes of soybean oil and cakes, domestic farm price of cottonseed and maize, world price of crude oil, population, and *GDP* are exogenous variables.

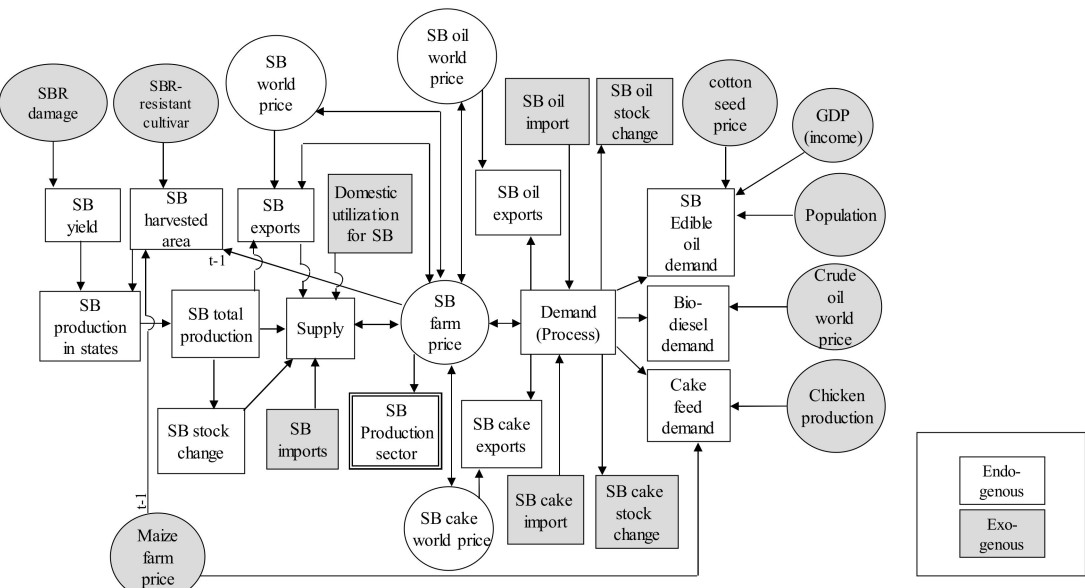

**Figure 1.** Flowchart of supply and demand of soybean econometric model in Brazil.

An augmented Dickey–Fuller (ADF) test was applied to examine the stationarity of all variables, except the yield functions. Johansen cointegration tests were applied in the next step if the variables

had unit roots. If the variables did not have cointegrating relations, then difference functions were applied, as described in Hung et al. [28].

### 2.2. Data

The data on *Y* and *A* for soybeans in each state or region of Brazil were derived from CONAB [20]. The following data were obtained from FAO-STAT [11]: *sbFP*, *sbEX*, *sbSTC*, *sbFO*, *sbSE*, *sbFE*, *sbIM*, *sbQ*, *sbPR*, *oilPR*, *oilEX*, *oilIM*, *oilSTC*, *oilED*, *oilBD*, *cakQ*, *cakEX*, *cakFE*, *cakIM*, *cakSTC*, *GDP*, and *POP*. The data on *CPI*, *sbWP*, *oilWP*, *cakWP*, and *chQ*, were acquired from the World Bank [29]. Due to hyperinflation, there were multiple currencies in Brazil, thus, *FP* (LCU/metric ton) was converted to USD/metric ton, using the exchange rate, which was derived from WB [29].

### 2.3. Scenario Setting

We set up five scenarios to simulate the effect of SBR-resistant cultivars on soybean production (Table 1). The base scenario was maintaining the current status. In Scenario 1, soybean production is not affected by SBR. In Scenarios 2 and 3, fungicide becomes ineffective and soybean production gradually decreases due to SBR in south and southeast Brazil and entire Brazil, respectively. In Scenarios 4 and 5, SBR-resistant cultivars are adopted in south and southeast Brazil and entire Brazil, respectively. The equilibrium domestic farm prices were estimated in each scenario.

**Table 1.** Scenario setting, assumptions in each scenario.

| | Setting | Assumptions |
|---|---|---|
| Baseline | Continuing current situation | Forecasting began in 2019. |
| Scenario 1 | No damage due to SBR | Recover the historical yield due to SBR and measure yield function. |
| Scenario 2 | Loss of soybean production in south and southeast regions of Brazil | Fungicide becomes ineffective (Equation (18)) and there is a production loss of 8.49% in south and southeast region. |
| Scenario 3 | Loss of soybean production in all the states of Brazil | Fungicide becomes ineffective (Equation (18)) and there is a production loss of 8.49% in all the states of Brazil. |
| Scenario 4 | SBR-resistant cultivars will be adopted under Scenario 2 | SBR-resistant cultivar will be diffused (Equation (19)) up to 60.9% of planted area in south and southeast region under Scenario 2. |
| Scenario 5 | SBR-resistant cultivars will be adopted under Scenario 3 | SBR-resistant cultivar will be diffused (Equation (19)) up to 60.9% of planted area in all the states of Brazil under Scenario 3. |

In Scenario 1, according to the information on soybean production loss, we divided the loss of production based on the proportion of planted area of states where damage was reported from 2001/2002 to 2013/2014 crop years [5]. We measured the yield functions by considering the proportion of damaged yield in each state or region. Subsequently, from the estimated yield and projected area of baseline analysis, we calculated the soybean production in Scenario 1. For Scenarios 2 and 3, we referred to the case of demethylation inhibitor (DMI), which is the most utilized fungicide in Brazil, but has a decreasing efficacy [30]. We applied the logarithmic function to describe the efficacy of DMI ($E_{DMI}$) as follows because the decrease in efficacy followed a curvilinear path and not a linear one:

$$E_{DMI}(t) = a_e + \frac{b_e - a_e}{1 + EXP[-c_e(t - d_e)]} + e_e t, \tag{18}$$

where $a_e$ and $b_e$ are the minimum and maximum levels of efficacy, respectively, $c_e$ is the efficacy reduction rate, $d_e$ is the inflection year, and $e_e$ is the slope of the time trend. The largest loss was

incurred in the 2003 to 2004 crop season in Brazil, with a loss of 8.49% of total soybean production [5]. The minimum level of efficacy in Equation (18) reflects the largest rate of loss in this year. For Scenarios 4 and 5, we referred to the case of genetically modified (GM) Intacta soybean cultivars in Brazil. In 2016, 94% of cultivars were GM soybean [31]. Out of these, 89% registered in 2017 were Intacta soybeans [32]. In 2013, the planted area of Intacta soybeans in Brazil was only 7.9%, but this increased to 60.9% in 2016. On the basis of the diffusion rate of Intacta soybeans, we estimated the diffusion area by the following logistic function [33]:

$$A_c(t) \;=\; a_{ci} + \frac{b_{ci} - a_{ci}}{1 + EXP[-c_{ci}(t - d_{ci})]} + e_{ci}t, \tag{19}$$

where, $i$ is the identification number of the state, $A_c$ is the planted area of SBR-resistant cultivars, $t$ is time (year), $a_c$ is area in Scenario 2, $b_c$ is the maximum planted area of SBR-resistant cultivars (60.9% of baseline area), $c_c$ is the diffusion rate, $d_c$ is the inflection year, and $e_c$ is the slope of the baseline area. We applied $e_c$: in only the case where baseline area increased. With respect to the case of Intacta soybeans, we assumed that the SBR-resistant cultivars are GM cultivars, spreading since 2017 and covering 3.89% of total planted area in each state or region [2]. The inflection point will be 2021 and it should have the same diffusion rate as Intacta. The planted area of SBR-resistant cultivars will be diffused up to 60.9% by 2023. The planted area of conventional cultivars was determined by subtracting the planted area of SBR-resistant cultivars from the planted area of soybeans in each state or region. The yield of SBR-resistant cultivars was assumed to be 3.11 metric ton/ha, which is equal to the SBR-resistant cultivar, TMG7067IPRO, that has already been released [34]. The same trend was considered for each state or region. The yield of conventional cultivars in Scenarios 4 and 5 was initially assumed to be the same as that in Scenarios 2 and 3, which then decreases with the decrease in the efficacy of fungicides in Scenarios 4 and 5. Production of SBR-resistant cultivars was examined by multiplying the yield by area of SBR-resistant cultivars.

## 3. Results

### 3.1. Baseline Analysis

The estimation period was 1981 to 2013, for which all data were available. The assumptions of the simulation were as follows (Figure A1): (1) The forecasted growth of *CPI* is the average annual growth between 2011 and 2013, (2) the forecasted value of real *GDP* is the annual average between 2011 and 2013, (3) the forecasted growth of population is the average annual growth between 2011 and 2013, and (4) the linear trend of the yield functions continues during the simulation period. The base year of the simulation of *Y*, *A*, *sbEX*, *STC*, *oilEXP*, *cakFE*, *cakEX*, and *sbFP* was 2019. The latest data for the calibration of *sbEX*, *sbSTC*, *cakFE*, *cakEX*, and *sbFP* was taken from the USDA [35]. The base year of the simulation of *oilED*, *oilBD*, *oilWP*, and *cakWP* was 2013. The forecasted period was from the base year untill 2030. Hence, the intercepts of the functions were calibrated to the latest data.

Tables 2 and 3 show the parameters of yield and planted area functions of soybeans in each state or region. The yield of soybeans depends on the time trend, denoted by a wide range of technological developments, such as improved irrigation systems. Some reports have indicated that the yield of soybeans in Brazil would increase if the water deficit and crop management systems were improved [36,37]. Our model included this potential increase in the yield of soybeans. The planted area was expanded when the farm price rose in the previous year in each state or region and it was affected by the previous crop year in a positive direction (Table 3). This indicates that rising farm prices encourage farmers to cultivate and expand the planted area. The production of soybeans, which depends on yield and planted area, will increase with an increase in yield (Figure 2). Figure 2 shows the simulation results of the production of soybean in the top three producing states and throughout Brazil. The production of soybeans in Mato Grosso, Paraná, and Rio Grande do Sul is expected to increase to 44.1 million, 23.8 million, and 20.4 million metric tons between 2019 and 2030,

respectively. The production of soybeans in Brazil will increase to 143.4 million metric tons during the forecasted period.

**Table 2.** Parameters with *t*-values of time trend and adjusted coefficient of determination (Adj. $R^2$) of yield function of soybeans in each state or region.

| State/Region | Trend | Adj. $R^2$ |
|---|---|---|
| North and northeast | 0.0403 *** (4.94) | 0.849 |
| Central-West | | |
| Distrito Federal | 0.0389 *** (7.79) | 0.855 |
| Goiás | 0.0403 *** (14.0) | 0.941 |
| Mato Grosso do Sul | 0.0326*** (6.41) | 0.586 |
| Mato Grosso | 0.0336 *** (11.4) | 0.961 |
| Southeast | | |
| Minas Gerais | 0.0381 *** (15.1) | 0.937 |
| São Paulo | 0.0324 *** (9.39) | 0.754 |
| South | | |
| Paraná | 0.0293 *** (5.03) | 0.660 |
| Rio Grande do Sul | 0.0250 *** (3.05) | 0.371 |
| Santa Catarina | 0.0377 *** (4.56) | 0.868 |

***, 1% significance level.

**Table 3.** Parameters with *t*-values of area and real farm price of soybean (sbRFP) and maize (maRFP), adjusted coefficient of determination (Adj. $R^2$), and the probabilities of the augmented Dickey–Fuller test (ADF pv.) of area function of soybeans in each state or region.

| State/Region | $Area_{t-1}$ | $sbRFP_{t-1}$ | $maRFP_{t-1}$ | Adj. $R^2$ | ADF pv |
|---|---|---|---|---|---|
| North and northeast | 0.909 *** (9.10) | 527.4 ** (2.23) | −402.1 (−1.39) | 0.987 | 0.999 (LC) |
| Central−West | | | | | |
| Distrito Federal | 0.239 (1.28) | 21.4 (1.23) | −20.8 (−1.10) | 0.146 | 0.274 (D) |
| Goiás | 0.177 (1.11) | 323.8 (1.23) | −372.2 (−1.45) | 0.303 | 0.153 (D) |
| Mato Grosso do Sul | 0.462 ** (2.70) | 773.9 ** (2.15) | −247.7 (−1.00) | 0.287 | 0.387 (D) |
| Mato Grosso | −0.0820 (−0.420) | 1066 (1.28) | | 0.163 | 0.819 (D) |
| Southeast | | | | | |
| Minas Gerais | 0.781 *** (7.00) | 302.8 ** (2.21) | −305.5 (−1.66) | 0.940 | 0.0287 (L) |
| São Paulo | 0.337 *** (3.90) | 186.7 ** (2.28) | −201.1 ** (−2.44) | 0.856 | 0.00712 (L) |
| South | | | | | |
| Paraná | 0.301 * (1.75) | 689.5 (1.48) | | 0.219 | 0.593 (D) |
| Rio Grande do Sul | 0.377 ** (2.57) | 769.1 (1.52) | −1089 * (−2.04) | 0.416 | 0.801 (D) |
| Santa Catarina | 0.258 (1.51) | 65.1 (1.04) | −70.2 (−1.17) | 0.145 | 0.986 (D) |

***, 1% significance level; **, 5% significance level; *, 10% significance level; ADF pv, provability values of the augmented Dickey–Fuller test; L, linear function is estimated; D, difference function is estimated; LC, linear function is estimated due to the existence of cointegration.

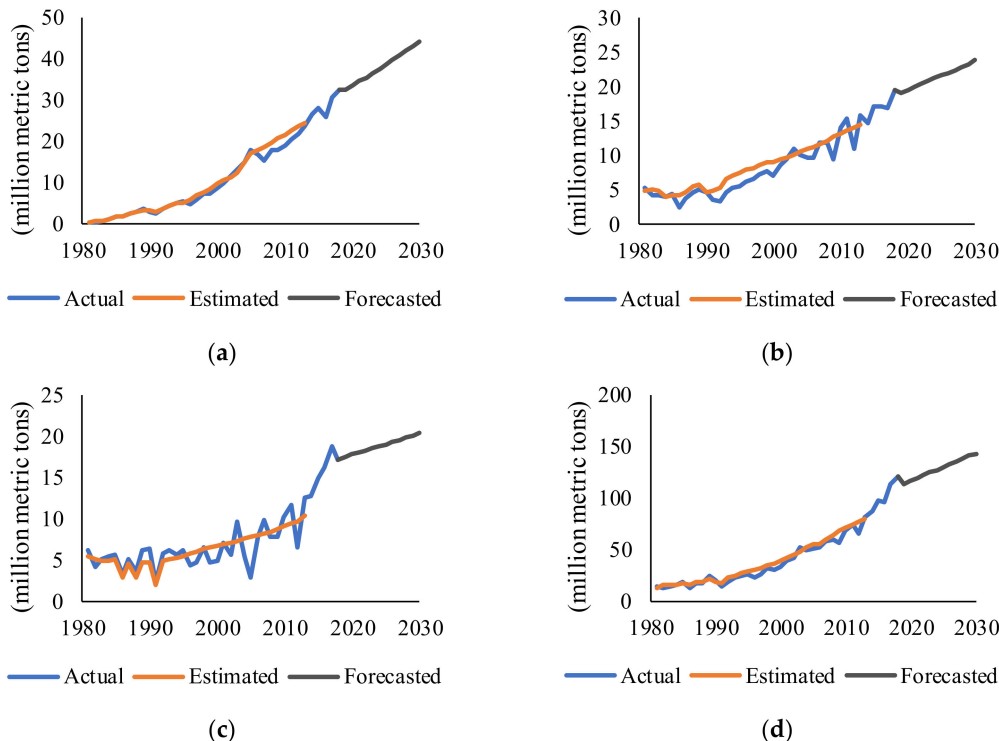

**Figure 2.** Production of soybeans in (**a**) Mato Grosso; (**b**) Paraná; (**c**) Rio Grande do Sul; and (**d**) entire Brazil.

Soybeans produced in Brazil are either exported or processed into soybean oil and cake. Soybean oil corresponds to approximately 20% of crushed soybeans. During the estimation period, 34% of soybean oil was exported (54% edible on average) [11]. Figure 3 shows edible soybean oil consumption per capita and soybean oil consumption for biodiesel. Figure 4 shows soybean cake consumption for feed. Soybean cake, corresponding to approximately 80% of crushed soybeans, was exported (67%) and used as feed for livestock (34%) during the estimated period [11]. Consumption of soybean cake for feed will increase due to a global shift towards more meat consumption, but the consumption of soybean oil will not increase as much [12]. Figure 5 shows the simulation result of the equilibrium farm price. The farm price is estimated to be 334.2 USD/metric ton in 2030.

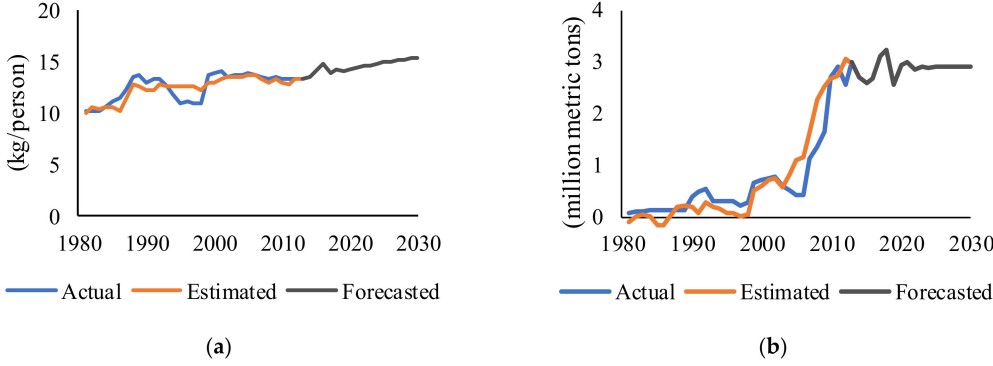

**Figure 3.** Consumption of edible soybeans per capita (**a**) and biodiesel (**b**) in Brazil.

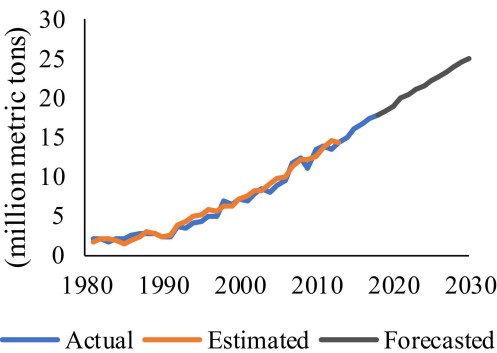

**Figure 4.** Consumption of cake feed in Brazil.

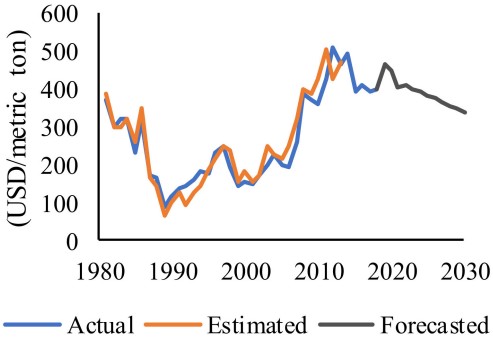

**Figure 5.** Farm price in Brazil.

### 3.2. Simulation Analysis

Figure 6 shows the outlook of the production of soybeans in Mato Grosso, Paraná, Rio Grande do Sul, and throughout Brazil in the five scenarios. The production of soybeans in Mato Grosso, Paraná, Rio Grande do Sul, and throughout Brazil is expected to be 44.5 million, 24.0 million, 21.4 million, and 146.8 million metric tons in 2030, respectively (Scenario 1). Although the estimated recovery of soybean production from SBR seems small, this small damage will affect the domestic farm price in Scenario 1. The domestic farm price is estimated to decrease due to an increase in soybean production in this scenario (Figure 7a). In Scenario 2, production loss in these areas will increase the domestic farm price (Figure 7b) as the soybean production decreases in south and southeast Brazil. In Scenario 3, the domestic farm price increases due to the reduction in the production of soybeans throughout Brazil (Figure 7a).

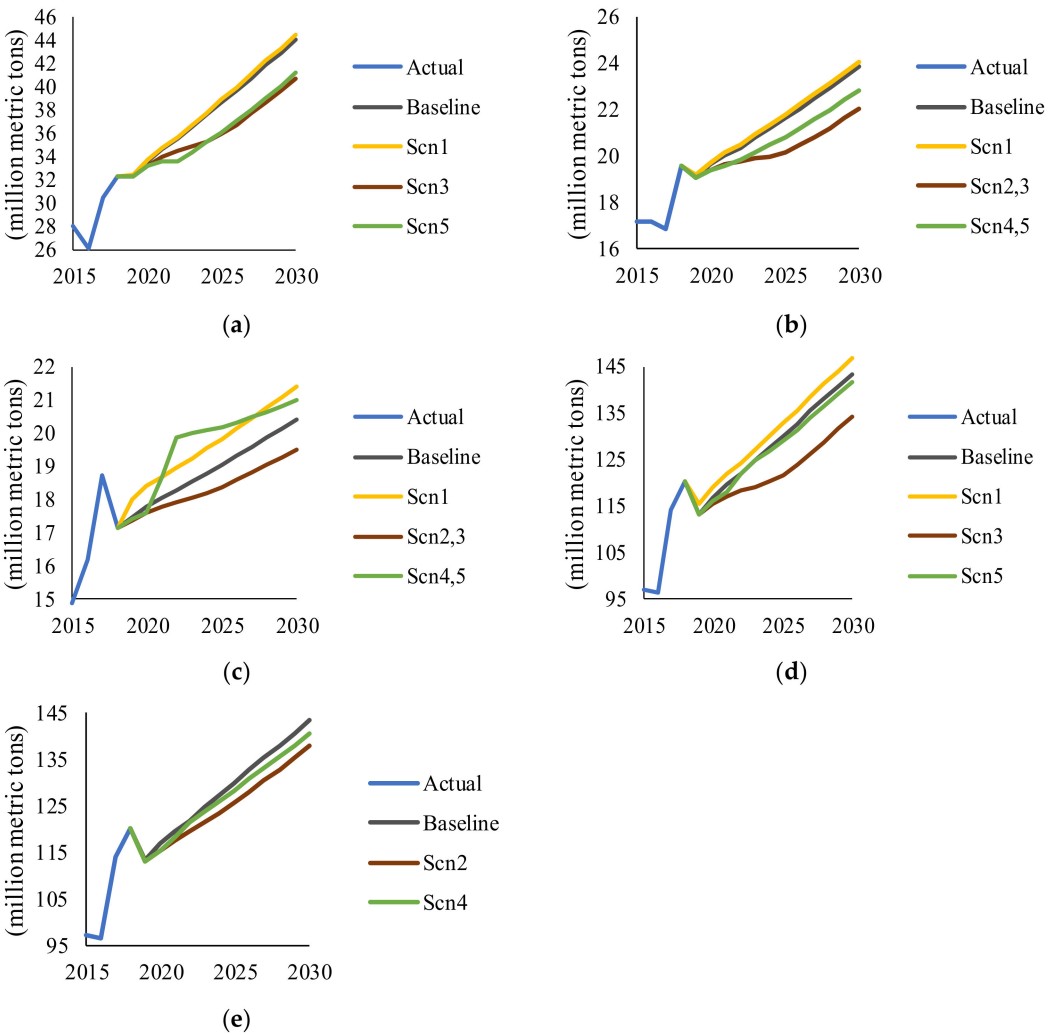

**Figure 6.** Forecast of soybean production in (**a**) Mato Grosso; (**b**) Paraná; and (**c**) Rio Grande do Sul; and (**d**,**e**) throughout Brazil in the five scenarios (Scenarios 1 to 5). As Scenario 2 and Scenario 4 in Mato Grosso had similar baselines, their results were not shown due to simplification.

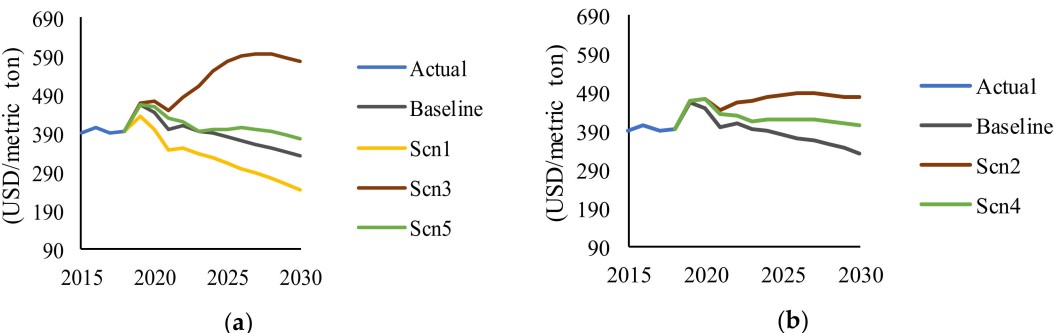

**Figure 7.** Forecast of farm price in the (**a**) Scenarios 1, 3, and 5; (**b**) Scenarios 2 and 4.

## 4. Discussion

Does adopting SBR-resistant cultivars help maintain a stable supply of soybeans? The yield in each state or region determines the total soybean production, which is also the total supply of soybeans. The yield of conventional cultivars in Mato Grosso was high, at 3.39 (metric ton/ha) in 2018, while that in Rio Grande do Sul was 3.01 (metric ton/ha) [20]. In this study, the yield of SBR-resistant cultivars is assumed to be 3.11 (metric ton/ha), based on Dorneles et al. [34]. Moreover, we considered

technological advancement based on trends in each state or region. When the SBR- resistant cultivar is disseminated in Mato Grosso, the yield of this state is projected to increase to 3.54 (metric ton/ha) by 2030, which is lower than that of the conventional cultivar at 3.74 (metric ton/ha) (Figure A2). In the case of Rio Grande do Sul, the yield of conventional cultivar was 3.01 (metric ton/ha) in 2018 [20] and disseminating SBR-resistant cultivars will increase the soybean production there. Disseminating SBR-resistant cultivars in the south and southeast regions will increase soybean production, thus, the domestic farm price will almost be the same as that at the baseline (Figure 7b). However, the regions where the yield of conventional cultivars is higher than that of SBR-resistant cultivars cannot alleviate the rise of domestic farm price of soybean to a large extent (Figure 7a). Thus, the extent of alleviation of rising domestic farm price due to the decrease in soybean production caused by SBR depends on the yield of conventional cultivars in each state or region. Researchers who breed SBR-resistant cultivars aim for similar yields without compromising the taste and appearance of conventional cultivars in each area [38]. Thus, there is a possibility that the results of our simulation analysis could be somewhat different. Moreover, we assumed that the dissemination of SBR-resistant cultivars was the same as the dissemination of Intacta cultivars as mentioned earlier. These assumptions are possible limitations of our estimation.

We estimated the cost-saving due to the adoption of SBR-resistant cultivars based on the equilibrium quantities in each scenario (Table 4). In Scenario 3, we assumed that fungicide is applied twice for SBR-resistant cultivars and four times for conventional cultivars in Scenario 5 according to Dorneles et al. [34]. The cost of fungicide was assumed as 25 USD/ha [5]. The cost-saving due to the adoption of SBR-resistant cultivars in Mato Grosso, Paraná, and Rio Grande do Sul are estimated to be 338 million, 185 million and 170 million USD, respectively. The cost-saving due to the adoption of SBR-resistant cultivars in Brazil is projected to reach 1.28 billion USD by 2030 (Table 4). The cost of applying fungicide in the 2013/2014 crop year was 2.2 billion USD countrywide [5]. Thus, adopting SBR-resistant cultivars could save almost half of the amount spent on SBR control.

**Table 4.** Estimated planted area, diffusion area of SBR-resistant cultivars, and area of conventional cultivars, application of fungicide when adopting SBR-resistant cultivars and conventional cultivars, total cost in Scenarios 5 and 3, and cost-saving in Mato Grosso (MT), Paraná (PR), Rio Grande do Sul (RS), and throughout Brazil in 2030.

| | MT | PR | RS | BRA |
|---|---|---|---|---|
| Planted area (million ha) | 11.1 | 6.06 | 5.59 | 35.8 |
| SBR-resistant cultivars area (million ha) | 6.76 | 3.69 | 3.40 | 21.8 |
| Conventional cultivars area (million ha) | 4.34 | 2.37 | 2.19 | 16.4 |
| Fungicide application to SBR-resistant cultivars (million USD) | 338 | 185 | 170 | 1280 |
| Fungicide application to conventional cultivars (million USD) | 434 | 237 | 219 | 1640 |
| Total cost of fungicide application in Scenario 5 (million USD) | 772 | 422 | 389 | 2920 |
| Total cost of fungicide application in Scenario 3 (million USD) | 1110 | 606 | 559 | 4200 |
| Cost-saving (Scenario 3 to Scenario 5) (million USD) | 338 | 185 | 170 | 1280 |

Fungicide was applied twice on the resistant cultivars and four times on the conventional cultivars [34]. The cost of fungicide is assumed to be USD25/ha/spray [5].

We proposed the worst case in Scenario 3, although it seems improbable. However, there are some factors other than SBR that reduce soybean production, such as climate change. There is an increase in droughts, especially in the northeast region, where rainfall has been below normal in the last two decades [39]. It is forecasted that drought will increase in this region, while rainfall will increase in the south and southeast regions [40]. As the growth of SBR is influenced by precipitation and moisture [41], there is a possibility of an epidemic in the south and southeast regions [13]. Caetano et al. [42] showed that cultivation of soybeans in the central region, which is the most productive area, will not remain environmentally suitable due to climate change. As our models do not consider climate change, further studies must be conducted to take into account the effect of climate change on both

soybean production and SBR in the future. Furthermore, our scenarios assume damage due to SBR, and not by climate change. However, production loss will actually occur because of climate change. If production throughout Brazil falls, domestic farm price will rise (Figure 7). As Brazil is one of the top soybean-producing and exporting countries in the global market, increase in domestic farm price will affect the world price of soybeans and the global market.

Scenarios 4 and 5 assume that the effect of SBR-resistant cultivars will last up to 2030. However, there is a possibility that SBR-resistant cultivars could be ineffective. In Brazil, herbicide-tolerant GM soybean cultivar, Roundup Ready, has been registered since 1999 [32], and herbicide-tolerant and soybean cultivars that are herbicide-tolerant and pesticide-resistant are being used in Brazil. Many pesticide-resistant GM soybean cultivar, Intacta, has been registered since 2012 [32]. As of 2019, GM Brazilian cultivars have been developed and registered each year [43]. The SBR-resistant cultivars in Scenarios 4 and 5 are assumed to correspond to several types of rust populations. Hence, Scenarios 4 and 5 would be realistic. As Brazil is a major soybean producing and exporting country [11], early and quick diffusion of SBR-resistant cultivars in Brazil is desirable to maintain stable and sustainable soybean production.

## 5. Conclusions

SBR is one of the most severe threats to soybean production in Brazil. If soybean production reduces, soybean oil and cake production will also fall. This would have a significant impact on the global soybean and soybean products market, as Brazil is a major player in this sector. To avoid significant loss due to SBR, several SBR-resistant cultivars have been developed. We analyzed the impact of SBR-resistant cultivars on soybean production in Brazil with a supply and demand model. The baseline analysis, which is used in subsequent scenarios, showed that soybean production will steadily increase due to the technological progress over the simulation period. This simulation was used as the starting point of the five scenarios. Scenario 1 assumed that soybean production was not affected by SBR. Scenarios 2 and 3 assumed that fungicide becomes ineffective and soybean will be damaged gradually by SBR in the south and southeast regions and all the states of Brazil. Scenarios 4 and 5 assumed SBR-resistant cultivar adoption in the south and southeast regions and all the states of Brazil. We compared the cost-saving on fungicide use by adopting SBR-resistant cultivars. Cost-saving by adopting SBR-resistant cultivars in Mato Gross, Paraná, and Rio Grande do Sul, and throughout Brazil will be 338 million, 185 million, 170 million, and 1.28 billion USD in 2030, respectively. If the environment becomes increasingly favorable for SBR due to global warming, there is a possibility that the worst-case scenario could happen. The adoption and early and quick diffusion of SBR-resistant cultivars will be needed for sustainable soybean production not only as a source of food, but also as biodiesel and feed for livestock.

**Author Contributions:** Conceptualization by Y.I.I. and J.F.; software by Y.I.I.; formal analysis by Y.I.I.; writing—original draft preparation by Y.I.I.; writing—review and editing by Y.I.I. and J.F.; and supervision by J.F. All authors have read and agreed to the published version of the manuscript.

**Funding:** This research received no external funding.

**Acknowledgments:** We would like to thank Naoki Yamanaka of the Japan International Research Center for Agricultural Sciences (JIRCAS) for providing the useful comments regarding the setting up of the scenarios in this paper.

**Conflicts of Interest:** The authors declare no conflict of interest.

# Appendix A

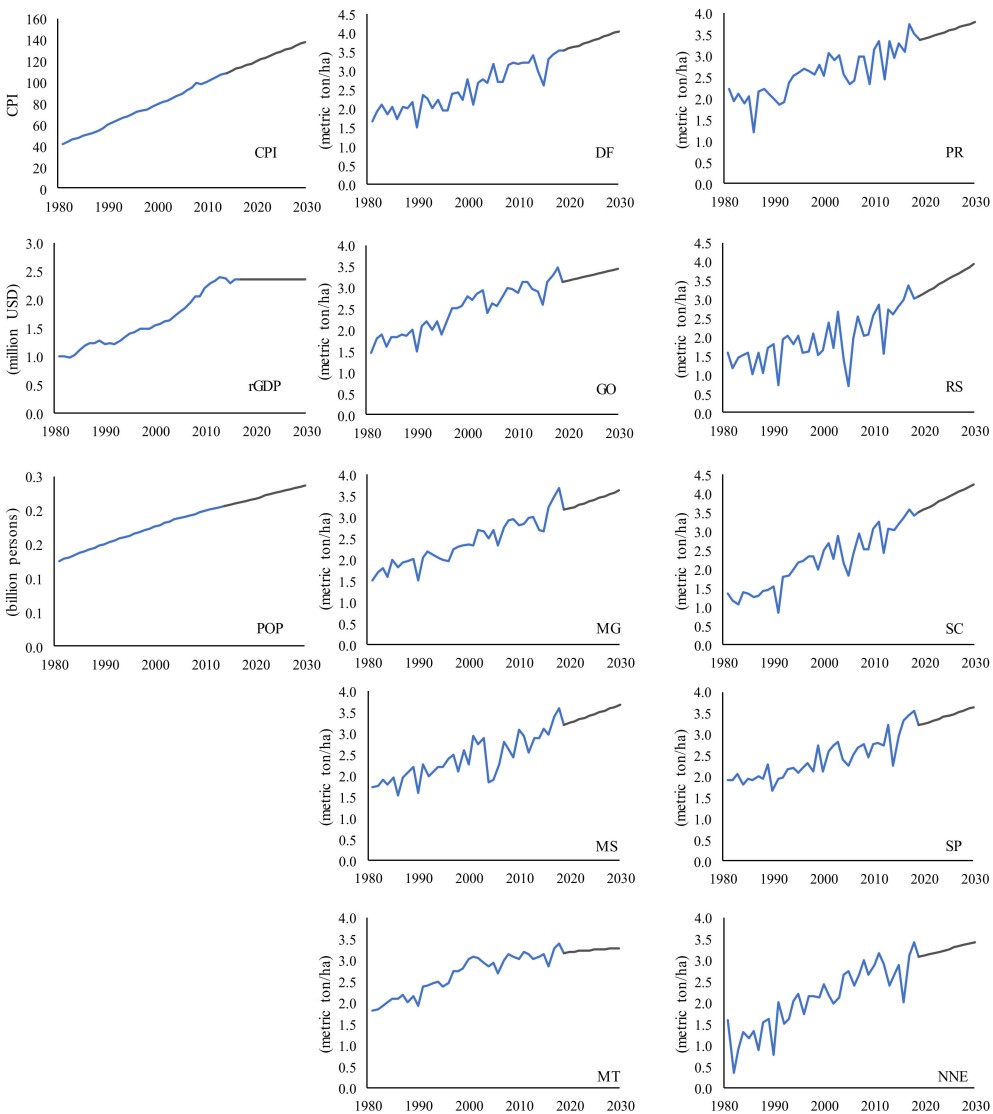

**Figure A1.** Assumptions of CPI, real GDP (rGDP), population (POP), and yields in Brazil. DF, Distrito Federal; GO, Goiás; MG, Minas Gerais; MS, Mato Grosso do Sul; MT, Mato Grosso; PR, Paraná; RS, Rio Grande do Sul; SC, Santa Catarina; SP, São Paulo; and NNE, north and northeast regions. The blue and gray lines indicate actual situation and assumptions, respectively.

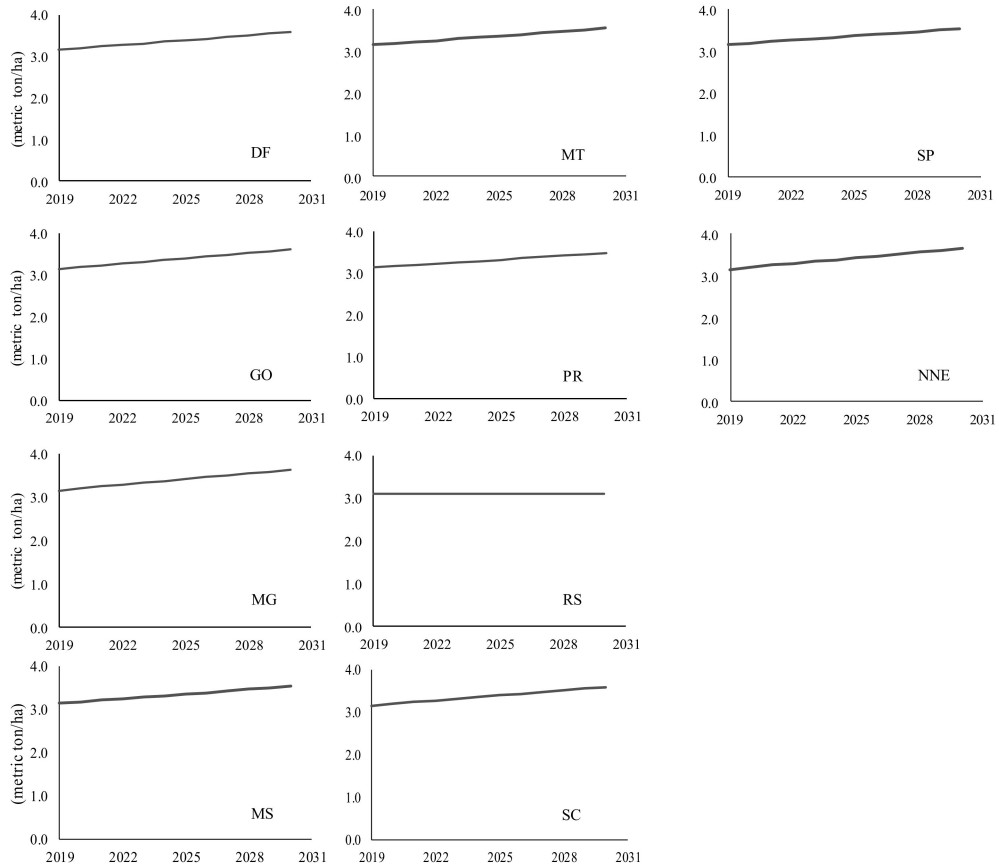

**Figure A2.** Assumptions of the yield of SBR-resistant cultivars in each states or region under Scenarios 4 and 5 in the period 2019 to 2030.

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
