# Peer review of "Evaluating the Contribution of Soybean Rust- Resistant Cultivars to Soybean Production and the Soybean Market in Brazil: A Supply and Demand Model Analysis"

_sustainability, doi:10.3390/su12041422_

Round 1
Reviewer 1 Report
The manuscript has been improved by the revisions.
Author Response
Our manuscript has been checked by a professional English editing service.
Reviewer 2 Report
Review of
“Evaluating the Contribution of Soybean Rust Resistant Cultivars to Soybean Production and the Soybean Market in Brazil: A Supply and Demand Model Analysis”
(sustainability-679820)
Summary
The authors made efforts to incorporate reviewers’ comments and suggestions. But there are still some issues not solved sufficiently.
Comments
For scenarios:
Can you please list the current situation for the baseline in Table 1 so that readers can know clearly where you start from?
Based on the increase in the adoption of the cultivar Intacta in 2013-2016, the authors assume that the adoption rate of SBR resistant cultivars will also reach the same rate in 4 years, but keep this rate for the rest of the projection period. Do you think the growth may probably continue and thus the adoption rate may be higher than 60.9% after a few years?
You added a discussion about the climate change in lines 55-74. I agree that the linkage to the climate change and sustainability can make your paper more fit to the journal. However, you do not consider the climate change in your model. Does it possibly become a limitation of your study?
You assume that Brazil is a price taker in the world soybean market. Given the fact that it is the second largest exporter in the world, please justify your assumption. In Figure 1, you indicate that supply and world price mutually affect each other. But in the model, you assume the world price as fixed even if the supply changes.
For Materials and Methods:
Export equation (eq. 4). Export should be also affected by the world price. Line 130, there is a typo. It should be soybean cake exports. Equation 10, please make sure that the base year of the CPI is 2005, same as that being used to calculate the real GDP. Demand function of soybean oil for biodiesel. I think world crude oil is a closer substitute than is cottonseed oil. And the price of soybean oil for biodiesel instead of the price of soybean should be included. Equation 18. If t is time (year), then the interpretation of ec is wrong.
For Results
Table 3, Area in the head of the table should be labelled as lagged area. Table 3. Please explain why higher maize price resulted in higher soybean area if maize competes for land with soybeans in Brazil. Was it because of new crop land cultivation? Or is there a different competing crop in different regions? Please use “estimated” instead of “simulated” in your figures as they are from your model estimation. Figures 2 and 6. Please indicate why these three states are selected. Figure 6e. If there is an 8.49% loss each year, how can scenario 2 has the same production as does the baseline? Lines 279-280. The loss rate of 0.183% conflicts with the loss rate of 8.49% in Table 1. Also, it should be “damage” instead of “recovery”. Figure 7a. What causes the big difference in price between baseline and Scn1 given the similar production shown in Figure 6d.
For Discussion
Given the fact that conventional cultivars have higher yield in Mato Grosso of than in Rio Grande do Sul, is it possible that the yield of SBR-resistant cultivars may also vary across regions? If yes, then it is not reasonable to assume the yield of 3.11 mt/ha for SBR-resistant cultivars the same across the regions at the start of the projection.
Author Response
Response to Reviewer 2 Comments
Thank you very much for thoughtful and constructive feedback regarding our manuscript. We have revised our manuscript accordingly. We provide a point-by-point response to the questions and comments below.
For scenarios
Point 1. Can you please list the current situation for the baseline in Table 1 so that readers can know clearly where you start from?
Point 1. We added the year when the forecast began in Table 1.
Point 2. Based on the increase in the adoption of the cultivar Intacta in 2013-2016, the authors assume that the adoption rate of SBR resistant cultivars will also reach the same rate in 4 years, but keep this rate for the rest of the projection period. Do you think the growth may probably continue and thus the adoption rate may be higher than 60.9% after a few years?
Point 2. Yes, as you pointed, there is a possibility to alter the adoption rate after a few years. However, reference is needed to set the scenario and we did not have any more data to incorporate into our scenario setting. Thus, we added the explanation that our assumption has a limitation in the discussion section.
Point 3. You added a discussion about the climate change in lines 55-74. I agree that the linkage to the climate change and sustainability can make your paper more fit to the journal. However, you do not consider the climate change in your model. Does it possibly become a limitation of your study?
Point 3. We added a discussion about the climate change because this was suggested by reviewer1. We understand that we did not consider climate change in our model. Thus, we outlined the limitation in the section on climate change.
Point 4. You assume that Brazil is a price taker in the world soybean market. Given the fact that it is the second largest exporter in the world, please justify your assumption. In Figure 1, you indicate that supply and world price mutually affect each other. But in the model, you assume the world price as fixed even if the supply changes.
Point 4. Thank you for your comment. Given that Brazil is not a price taker, we changed the flow chart in Figure 1.
For Materials and Methods
Point 5. Export equation (eq. 4). Export should be also affected by the world price.
Point 5. We included the soybean world price in the export function (eq. 4). We also added the price linkage function of soybean price in eq. 15.
Point 6. Line 130, there is a typo. It should be soybean cake exports.
Point 6. Thank you for highlighting this error, we have revised this accordingly.
Point 7. Equation 10, please make sure that the base year of the CPI is 2005, same as that being used to calculate the real GDP.
Point 7. We added the base year of the CPI. We also changed the base year of real GDP from 2005 to 2010 and recalculated it accordingly.
Point 8. Demand function of soybean oil for biodiesel. I think world crude oil is a closer substitute than is cottonseed oil. And the price of soybean oil for biodiesel instead of the price of soybean should be included.
Point 8. Thank you for your suggestion. We included the crude oil world price instead of the price of cottonseed oil. We cannot obtain any data on the price of soybean oil for biodiesel; thus, we cannot include such data.
Point 9. Equation 18. If t is time (year), then the interpretation of ec is wrong.
Point 9. In equation 19 (revised version), t is time (year). However, in the case of area increase, the area of SBR-resistant cultivars must be increased as the same rate as that of conventional cultivars. We added an explanation of ec.
For Results
Point 10. Table 3, Area in the head of the table should be labelled as lagged area.
Point 10. We changed the heading of the table.
Point 11. Table 3. Please explain why higher maize price resulted in higher soybean area if maize competes for land with soybeans in Brazil. Was it because of new crop land cultivation? Or is there a different competing crop in different regions?
Point 11. Thank you for your comment. We changed maRFPt to maRFPt-1 and remeasured all area functions in eq. (12).
Point 12. Please use “estimated” instead of “simulated” in your figures as they are from your model estimation.
Point 12. We changed “simulated” into “estimated” in Figures 2-5.
Point 13. Figures 2 and 6. Please indicate why these three states are selected.
Point 13. We added explanation of why these three states are selected in the results section.
Point 14. Figure 6e. If there is an 8.49% loss each year, how can scenario 2 has the same production as does the baseline?
Point 14. In scenario 2, production is not the same as the baseline, but it was difficult to see because of our choice of color. We have revised this to enhance readability and clarity.
Point 15. Lines 279-280. The loss rate of 0.183% conflicts with the loss rate of 8.49% in Table 1.
Point 15. We have deleted these lines.
Point 16. Also, it should be “damage” instead of “recovery”.
Point 16. We have revised this accordingly.
Point 17. Figure 7a. What causes the big difference in price between baseline and Scn1 given the similar production shown in Figure 6d.
Point 17. Thank you for your comment. The way to converge the previous scenario 1 was different from that of other scenarios. We have recalculated and revised all the figures and results related to scenario 1.
For Discussion
Point 18. Given the fact that conventional cultivars have higher yield in Mato Grosso of than in Rio Grande do Sul, is it possible that the yield of SBR-resistant cultivars may also vary across regions? If yes, then it is not reasonable to assume the yield of 3.11 mt/ha for SBR-resistant cultivars the same across the regions at the start of the projection.
Point 18. The yield of SBR-resistant cultivars may vary in each state. Thus, our assumption regarding such cultivars may not be reasonable. However, there is a limitation regarding the reference. We added the limitation of our assumptions in the discussion section.

This manuscript is a resubmission of an earlier submission. The following is a list of the peer review reports and author responses from that submission.
Round 1
Reviewer 1 Report
Summary
The study aims to evaluate the effect of soybean rust (SBR) resistant cultivars on soybean production and soybean market in Brazil. Starting with a supply and demand model, the study estimates changes in soybean markets under three scenarios.
Comments
In general, the study lacks clear explanation of the purpose of the scenarios, detailed information for estimation methods, and a discussion of implications of results. In addition, the model is not robust and some assumptions need justifications.
For scenarios:
I do not understand why the authors set up such three scenarios. The base scenario is the current status. Scenario 1 assumes soybean production was not affected by SBR. Scenario 2 assumes the same damage loss to soybean crops by SBR as that during 2001-2004. Scenario 3 assumes SBR resistant cultivar adoption in all states of Brazil. What is the purpose of each scenario? Since the authors also assume the same adoption rate of GM cultivar for scenario 3 as that having occurred during 2013-2016, then what is the difference between the base scenario and the scenario 3? In the results, the authors show that the farm price will be similar in Scenario 3 and the base scenario, which to some degree coincides with my question here. In addition, do you keep the same proportion of damage all the way till 2030 for scenario 2? Is it possible the proportion of damage will be bigger if the contamination expands?
For model:
(1) If you are estimating a function, you need to add an error term in the equation as the covariates can’t fully explain the dependent variable.
(2) In Brazil, is sugar or corn an important competitive crop to soybean for land? If yes, please consider including the price(s) of competing crops in eq.2.
(3) Export (eq.4) is not determined solely by production. It also depends on domestic price and world price.
(4) The stock change (eq. 5) is also affected by changes in domestic demand and trade.
(5) What does “supply identity” mean? Based on eq.6, it is zero. Also, it is denoted the same as the processing identity (eq.7).
(6) Why does the export function of soybean cake (eq. 9) includes production while that of soybean oil (eq. 8) does not?
(7) Are there main substitutes for soybean oil in consumption in Brazil? If yes, then demand function of soybean edible oil (eq.10) should include prices of substitutes (such as corn oil, canola oil, and peanut oil, etc.). The same is applied to the demand function of soybean oil for biodiesel (eq. 11) and the demand function of cake feed (eq. 12).
(8) I suggest using the number of chicken and livestock instead of chicken price in eq.12.
For Figures:
(1) figure 1 is not well graphed and explained. The arrow direction mostly indicates influence. But there are several that do not make sense. For example, the arrows to SB oil production suggest import, stock change, and import all affect oil production, which is not true. The same is applied to the arrow from production to demand for both oil and cake, and arrows to SB cake production.
(2) Texts in figures 2-7 are all mirrored.
For methods and results:
(1) How did you conduct the projections for 2019-2030? Did you base them on the supply and demand model? Then how did you project world price that is used for projections of domestic price, soybean and cake consumptions?
(2) Lines 190-194 authors assume that the SBR resistant cultivars “spread from 2017 covering 1% of total planted area in each state because the SBR resistant cultivars…..were not yet popular”. But Lines 178-179 say that the cultivar of Intacta had increased from 7.9% to 60.9% during 2013 to 2016. Therefore, the assumed starting point of 1% in 2017 is not correct.
(3) Lines 195-196, since there already is SBR resistant cultivars, you should be able to check the yield difference between SBR resistant cultivars and conventional cultivars instead of making such an assumption.
(4) Lines 204-208, the assumptions need to be justified. You can graph those growth values for the period of 1981-2013 and see if your assumptions are reasonable.
(5) Please justify your assumptions in linear 261-262. This is an important assumption as the estimated cost is based on this assumption.

Reviewer 2 Report
The manuscript straightforwardly estimates the benefits of adopting soybean rust (SBR) resistant cultivars and finds that adoption of SBR resistant cultivars in Brazil is desirable for sustainability of soybean production. In the Sustainability journal aims and scope, the manuscript fits under the categories: “Implementation and monitoring of policies for sustainable development,” and “changing consumption and production patterns.” However there are several other factors affecting sustainability, e.g., climate change, conversion to soybean fields of land originally used for other purposes, and changes to resource usage, for example, that beg to be considered, particularly with respect to the Brazilian economy. Is SBR the most important of the many factors affecting sustainability of soybean production in Brazil? Perhaps, but it is not so obvious to the casual reader. Because this is a sustainability journal, it seems appropriate to place the results of the study in better context with other sustainability factors. Also, it is easier for the reader to follow manuscripts that separate results and discussion into separate sections. For these reasons, the manuscript would benefit significantly from consideration of the approximate economic costs and benefits of several other factors associated with the recent rapid growth of soybean production in Brazil in a separate discussion.
Here are some specific comments.
Line 85, 88, and possibly other places --Because the north and northeast states were grouped together into regions, it would be more precise to indicate these are growing regions rather than states. Indeed, because this is a journal about sustainability, it may be worthwhile to discuss if the effects are altered significantly by inclusion only of the top 5 states, for example. Climate change may be of concern, and it is addressed in general, but climate change may affect different states differently. Also, sustainability may be affected by changes in which regions grow the majority of the soybeans.
Line 119-120 – It may be worthwhile to add the conversion from tons, pounds, etc. to metric units.
In Figures 2-6, text lettering is reversed.
Line 204 – It may be worthwhile to include a table that lists all of the assumptions of the model and cultivar characteristics for the 3 scenarios.